# TASK-TO-INSTANCE PROMPT LEARNING
# FOR VISION-LANGUAGE MODELS AT TEST TIME

## ABSTRACT

Prompt learning has been recently introduced into the adaption of pre-trained vision-language models (VLMs) by tuning a set of trainable tokens to replace hand-crafted text templates. Despite the encouraging results achieved, existing methods largely rely on extra annotated data for training. In this paper, we investigate a more realistic scenario, where only the unlabeled test data is available. Existing test-time prompt learning methods often separately learn a prompt for each test sample. However, relying solely on a single sample heavily limits the performance of the learned prompts, as it neglects the task-level knowledge that can be gained from multiple samples. To that end, we propose a novel test-time prompt learning method of VLMs, called **T**ask-to-**I**nstance **P**rom**P**t **LE**arning (TIPPLE), which adopts a two-stage training strategy to leverage both task- and instance-level knowledge. Specifically, we reformulate the effective online pseudo-labeling paradigm along with two tailored components: an auxiliary text classification task and a diversity regularization term, to serve the task-oriented prompt learning. After that, the learned task-level prompt is further combined with a tunable residual for each test sample to integrate with instance-level knowledge. We demonstrate the superior performance of TIPPLE on 15 downstream datasets, *e.g.*, the average improvement of 1.87% over the state-of-the-art method, using ViT-B/16 visual backbone.

## 1 INTRODUCTION

Large-scale pre-trained vision-language models (VLMs), *e.g.*, CLIP (Radford et al., 2021), have shown impressive performance on diverse downstream tasks in the zero-shot evaluation. The ability of VLMs can be further advanced by prompt engineering (Liu & Chilton, 2022), which designs customized prompts that better describe the applied environments. However, prompt engineering may require expertise in the target task or domain and significant trial-and-error experimentation based on a held-out validation set, which makes it impractical.

Prompt learning can be an effective solution to overcome the challenges of prompt engineering (Zhou et al., 2022b). This is often achieved by fine-tuning the pre-trained model on a task-specific dataset, with the prompt as a set of trainable parameters. CoOp (Zhou et al., 2022b) is a representative method that proposes to learn prompts for downstream tasks in a supervised manner. Despite the promising performance, the supervised nature of CoOp (Zhou et al., 2022b) (Figure 1 (a)) and some follow-up works (Lu et al., 2022; Khattak et al., 2023; Chen et al., 2023; Zhou et al., 2022a; Bulat & Tzimiropoulos, 2023; Zhu et al., 2022) requires annotated data for training, which limits their applicability in the scenario where the labeled training data is inaccessible.

In this paper, we overcome the aforementioned challenges by investigating test-time prompt learning for VLMs, which aims to learn tailored prompts for downstream tasks using only unlabeled data during testing. Recently, TPT (Shu et al., 2022) and DiffTPT (Feng et al., 2023) have been proposed for addressing this task. Specifically, they separately learn a prompt for each unlabeled test sample by encouraging consistent predictions across different augmented views of a test sample. However, they are typically designed for adapting a VLM on a single instance only (Figure 1 (b)) while overlooking the task-level knowledge that can be acquired from multiple samples, which is essential for prompt learning on a specific downstream task.

To leverage both task- and instance-level knowledge, we propose a novel test-time prompt learning method called **T**ask-to-**I**nstance **P**rom**P**t **LE**arning (TIPPLE). Specifically, our TIPPLE adopts a

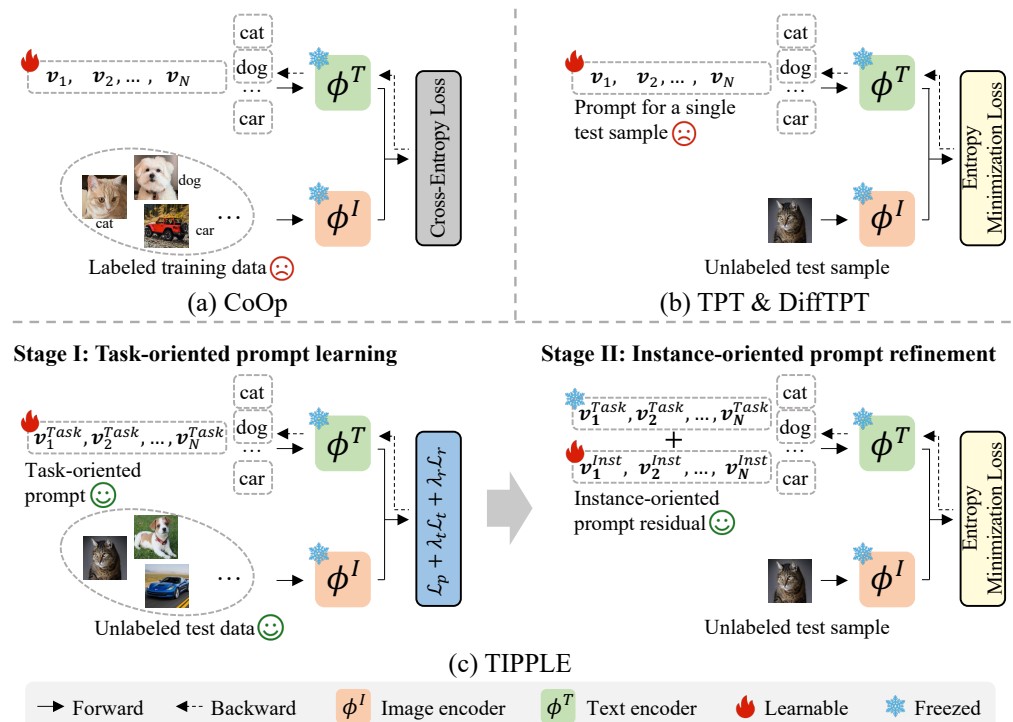

Figure 1: **Illustrating the differences among various methods.** (a) CoOp (Zhou et al., 2022b) needs labeled data to learn the prompt. (b) TPT (Shu et al., 2022) and DiffTPT (Feng et al., 2023) learn the prompt with a single unlabeled test sample at test time. (c) Our TIPPLE incorporates both task- and instance-level knowledge for test-time prompt learning in a two-stage training manner.

two-stage training strategy as shown in Figure 1 (c). In the first stage, TIPPLE is trained on the unlabeled test datasetwith visual and textual supervision. The visual supervision information is from the online pseudo-labels of confident test samples which are progressively generated using the latest learned prompt. The textual supervision is based on an auxiliary text classification task, which uses the trainable prompt to classify the class-related textual descriptions. In contrast to visual supervision which may contain noisy pseudo-labels, the textual supervision is noise-free since the textual descriptions are created by the templates and the class names. Besides, to further prevent the model from blindly trusting the pseudo-labels, we use a regularization term to diversify the model predictions. The above designs enable TIPPLE to learn the task-oriented prompt, containing rich task-level knowledge. In the second stage, utilizing the entropy minimization suggested in TPT, we tune a residual for each test sample based on the learned task-oriented prompt to integrate both task- and instance-level knowledge, instead of the hand-craft template initialization used in TPT.

Extensive experiments show that a new state-of-the-art performance is achieved by our method. In addition to the superior performance, we also verify that our task-oriented prompt learned on ImageNet is highly transferable to other datasets. Finally, the unique design of task-oriented prompt enables its effortless extension towards two additional scenarios – online streaming data and unlabeled training data, where previous methods are not applicable.

In summary, we list our contributions as follows:

- This study focuses on a practical yet under-studied research topic, test-time prompt learning of VLMs, and observes the limitation of existing methods, *i.e.*, it adapts the VLMs to a single test sample without considering task-level knowledge.

- We propose a novel test-time prompt learning method, TIPPLE, which leverages both task- and instance-level knowledge in a two-stage training manner. Building upon the pseudo-labeling paradigm, we exploit the task-level knowledge with two proposed novel components: textual supervision and diversity regularization.

- We evaluate the effectiveness of TIPPLE on 15 datasets covering diverse image classification tasks. The results demonstrate the state-of-the-art performance of our TIPPLE, often outperforming previous methods by large margins.
- In the cross-task setting and two scenarios with online streaming data and unlabeled training data, our method shows superior performance, further indicating its utility and scalability.

## 2 RELATED WORK

**Vision-language models.** Recently, VLMs which typically consist of an image encoder (*e.g.*, a CNN like ResNet-50 (He et al., 2016) or a vision transformer like ViT-B/16 (Dosovitskiy et al., 2020) ) and a transformer-based text encoder, learn visual representations from the supervision of natural language (Chen et al., 2020; Jia et al., 2021; Yuan et al., 2021; Radford et al., 2021; Li et al., 2022; Singh et al., 2022). The large-scale VLMs employ the contrastive training on large corpora of image-text pairs, *e.g.*, 400M in (Radford et al., 2021) and 1.8B in (Jia et al., 2021), and demonstrate an impressive transferable ability to the downstream tasks under few-shot and zero-shot settings. The success of VLMs on the recognition tasks motivates more studies on diverse downstream tasks, including dense prediction (Rao et al., 2022), video action recognition (Wang et al., 2021b), point cloud recognition (Zhang et al., 2022b), *etc*. In this paper, we study the downstream image recognition tasks using the most representative pre-trained model, CLIP (Radford et al., 2021).

**Prompt learning.** As an alternative to full fine-tuning and linear probing, prompt learning is first proposed to exploit pre-trained language models (Shin et al., 2020; Zhong et al., 2021). It fixes all pre-trained parameters but learns continuous vectors in the word embedding space. CoOp (Zhou et al., 2022b) firstly extends prompt learning to VLMs in the few-shot setting. Some subsequent works improve the performance of CoOp with prompt distribution learning (Lu et al., 2022), multi-modal prompt learning (Khattak et al., 2023), and optimal transport distance based optimization (Chen et al., 2023). Besides, CoCoOp (Zhou et al., 2022a) and follow-up works (Bulat & Tzimiropoulos, 2023; Zhu et al., 2022) focus on improving the generalization capability of learned prompts to novel classes. However, the aforementioned methods require labeled data for training, which makes them inapplicable to the scenario without annotations. The recent work, TPT (Shu et al., 2022) and DiffTPT (Feng et al., 2023), inspires us to learn prompts on test samples to overcome this problem.

**Test-time learning.** Test-time learning (Liang et al., 2020; Kundu et al., 2020; Li et al., 2020; Sun et al., 2020; Schneider et al., 2020; Zhang et al., 2022a) aims to adapt a pre-trained model to unlabeled data during testing. Since the labels are unavailable, one challenge is to introduce high-quality supervision information. In previous works, consistency regularization (Yang et al., 2021b; Sun et al., 2022; Fleuret et al., 2021; Peng et al., 2022; Chen et al., 2022a) and entropy minimization (Sohn et al., 2020; Li et al., 2020; Xia et al., 2022; You et al., 2021; Yan et al., 2021; Fleuret et al., 2021) are commonly-used optimization objectives. Moreover, the cross-entropy loss based on pseudo-labels (Liang et al., 2020; 2021; Qu et al., 2022; Yang et al., 2021a; Chen et al., 2022b) shows the superiority to solve this problem, and thus how to generate accurate pseudo-labels becomes an important research topic. Different from these works for vision models, we explore both visual and textual supervision information for adapting VLMs. Instead of updating the entire model, some works optimize part of model parameters such as the batch normalization layers (Wang et al., 2021a) and the feature extractor (Liang et al., 2020). In the context of VLMs, we follow TPT (Shu et al., 2022) to fix all pre-trained parameters and only optimize textual prompts at test time, but our method is designed for leveraging task-specific knowledge.

## 3 TASK-TO-INSTANCE PROMPT LEARNING

### 3.1 PRELIMINARIES

**A revisit of CLIP.** The image and text encoders of CLIP are denoted as $\phi^I(\cdot)$ and $\phi^T(\cdot)$, respectively. The image encoder transforms a given image $x \in \mathbb{R}^{H \times W \times C}$ into a feature vector $z = \phi^I(x) \in \mathbb{R}^D$, where $H, W$, and $C$ represent the image height, width, and the number of channels, respectively, and $D$ is the feature dimension. The text encoder generates features for a sequence of word tokens. The aligned features of the image and text enable us to perform zero-shot image recognition using CLIP.

Given $K$ classes, CLIP can output the prediction probability of the input image $x$ during testing. The prediction process is based on the class-specific text inputs which are formed by the prompt

template and the class names, *e.g.*, "a photo of a {class}". By calculating the similarities between the image feature $z$ and the text features $\{q_i\}_{i=1}^K$, where $q_i \in \mathbb{R}^D$ denotes the text feature of the class-specific text input, one can obtain the prediction probability of $x$ with respect to the class $y_i$ ($i \in \{1, 2, ..., K\}$), which can be formulated as:

$$p(y_i|\boldsymbol{x}) = \frac{\exp(\text{sim}(\boldsymbol{z}, \boldsymbol{q}_i)/\tau)}{\sum_{j=1}^K \exp(\text{sim}(\boldsymbol{z}, \boldsymbol{q}_j)/\tau))}, \tag{1}$$

where $\text{sim}(\cdot, \cdot)$ indicates the cosine similarity and $\tau$ is the temperature coefficient of CLIP.

**Prompt learning on the labeled training data.** Inspired by the success of prompt learning in natural language processing, recent works (Zhou et al., 2022b;a) introduce this idea to advance VLMs. Instead of using hand-crafted prompts, such as "a photo of a", they learn the continuous prompt with the labeled training data on the downstream tasks. $\boldsymbol{V} = [\boldsymbol{v}_1, \boldsymbol{v}_2, ..., \boldsymbol{v}_N]^\top \in \mathbb{R}^{N \times M}$ denotes the learnable prompt containing $N$ context tokens and $\boldsymbol{c}_i \in \mathbb{R}^M$ ($i \in \{1, 2, ..., K\}$) is the word embedding vector of the name of the $i$-th class $y^i$, where $M$ is the dimension of the word embedding vector (*e.g.*, 512 for CLIP). The prediction probability distribution of $\boldsymbol{x}$ with the prompt $\boldsymbol{V}$ is $\Phi_{\boldsymbol{V}}(\boldsymbol{x}) = [p(y_1|\boldsymbol{x}), p(y_2|\boldsymbol{x}), ..., p(y_K|\boldsymbol{x})]^\top \in \mathbb{R}^K$, where $p(y_i|\boldsymbol{x}) = \frac{\exp(\text{sim}(\boldsymbol{z}, \phi^T(\{\boldsymbol{V}, \boldsymbol{c}_i\}))/\tau)}{\sum_{j=1}^K \exp(\text{sim}(\boldsymbol{z}, \phi^T(\{\boldsymbol{V}, \boldsymbol{c}_j\}))/\tau))}$. With the labeled training data $\mathcal{D}_{train}$ and the cross-entropy loss $\ell$, the prompt can be optimized in a supervised manner, as follows:

$$\boldsymbol{V}^* = \arg\min_{\boldsymbol{V}} \sum_{(\boldsymbol{x}, y) \in \mathcal{D}_{train}} \ell(\Phi_{\boldsymbol{V}}(\boldsymbol{x}_i), y_i). \tag{2}$$

**Test-time prompt tuning (TPT) (Shu et al., 2022).** To overcome the limitation of depending on labeled training data in previous prompt learning methods, TPT learns a prompt for each test sample. Given a test sample $\boldsymbol{x}_{test}$, it utilizes a family of augmentation functions to generate $S$ randomly augmented views $\mathcal{A}_i(\boldsymbol{x}_{test})(i \in \{1, 2, ..., S\})$. The prompt is obtained by minimizing the entropy of the averaged prediction probability distribution on the selected confident samples:

$$\boldsymbol{V}^* = \arg\min_{\boldsymbol{V}} \mathbf{H}(\tilde{\Phi}_{\boldsymbol{V}}(\boldsymbol{x}_{test})),$$

$$\text{where } \tilde{\Phi}_{\boldsymbol{V}}(\boldsymbol{x}_{test}) = \frac{1}{\rho S} \sum_{i=1}^S \mathbb{I}_{[\mathbf{H}(\Phi_{\boldsymbol{V}}(\mathcal{A}_i(\boldsymbol{x}_{test}))) < \xi]} \Phi_{\boldsymbol{V}}(\mathcal{A}_i(\boldsymbol{x}_{test})), \tag{3}$$

where $\mathbf{H}$ calculates the self-entropy of the prediction probability distribution, and the indicator function $\mathbb{I}_{[\mathbf{H}(\Phi_{\boldsymbol{V}}(\mathcal{A}_i(\boldsymbol{x}_{test}))) < \xi]}$ selects $\rho$ percent of confident samples using a cutoff threshold $\xi$.

## 3.2 TASK-ORIENTED PROMPT LEARNING

Different from TPT which separately learns a prompt for each test sample, TIPPLE learns the task-oriented prompt in the first stage. To achieve this, we design three key components, namely, online pseudo-label supervision, textual supervision, and diversity regularization, as shown in Figure 2.

**Online pseudo-label supervision.** To leverage the unlabeled test data, we propose to learn the prompt from the pseudo-label supervision. Instead of updating pseudo-labels after a fixed training period (Caron et al., 2018; Liang et al., 2020), we adopt an online method, which progressively utilizes the latest version of the learned prompt to generate more accurate pseudo-labels. To reduce the effect of false pseudo-labels, we filter the potential noisy pseudo-labels by discarding the test samples which have small confidence scores. Let $\mathcal{D}_{test}$ and $\mathcal{B}_{test} \subset \mathcal{D}_{test}$ denote the unlabeled test dataset and a randomly sampled batch of test samples, respectively. The loss function using the online pseudo-labels is formulated as:

$$\mathcal{L}_p = \frac{1}{Q} \sum_{\boldsymbol{x} \in \mathcal{B}_{test}} \mathbb{I}_{[\max(\Phi_{\boldsymbol{V}}(\boldsymbol{x})) > \eta]} \ell(\Phi_{\boldsymbol{V}}(\mathcal{A}(\boldsymbol{x})), \hat{y}), \tag{4}$$

where $\hat{y} = \arg\max \Phi_{\boldsymbol{V}}(\boldsymbol{x})$ denotes the pseudo-label. The indicator function $\mathbb{I}_{[\max(\Phi_{\boldsymbol{V}}(\boldsymbol{x})) > \eta]}$ is 1 if the confidence score is greater than $\eta$; otherwise 0. $Q = \sum_{\boldsymbol{x} \in \mathcal{B}_{test}} \mathbb{I}_{[\max(\Phi((\boldsymbol{x}))) > \eta]}$ is the number of selected samples in $\mathcal{B}_{test}$. For more effectively exploiting the selected samples, we apply strong data augmentations to $\boldsymbol{x}$ when optimizing the prompt, denoted as $\mathcal{A}(\boldsymbol{x})$. However, when generating the pseudo-labels, for more accurate prediction, they are disabled.

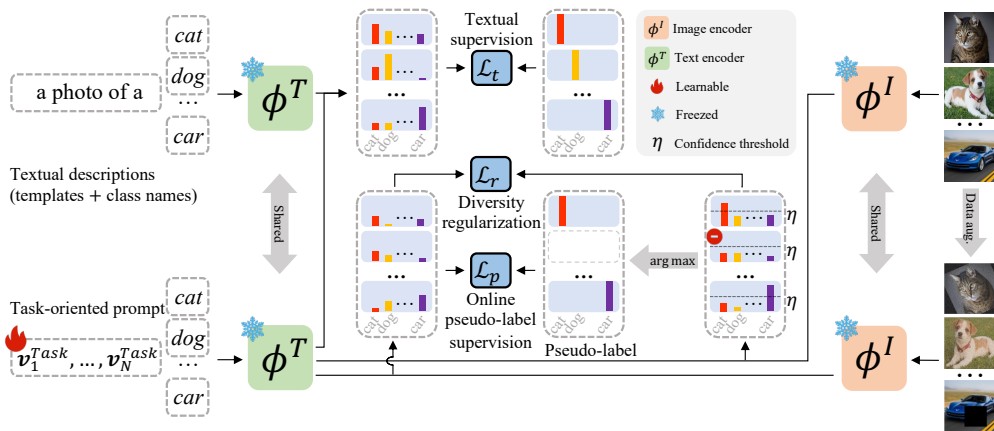

Figure 2: **Framework of the first stage of TIPPLE (task-oriented prompt learning).** There are three key components: the online pseudo-labels which supervise the predictions of the strongly-augmented samples selected using the confidence threshold $\eta$, the textual supervision which encourages correct predictions of class-related textual descriptions, and the diversity regularization which diversifies the model predictions.

**Textual supervision.** We further overcome the challenge of unavailable labeled data by utilizing an auxiliary text classification task, which provides more supervision information to train the prompt. Specifically, we first construct a text classifier based on the text encoder $\phi^T(\cdot)$ and the prompt $V$. We encourage it to correctly predict the ground-truth label of the textual description formed by a textual template (*e.g.*, "a photo of a {class}") and a class name, where the ground-truth label is assigned according to the class name. Given a textual description containing $L$ words whose embedding is $t \in \mathbb{R}^{L \times M}$, the text classifier predicts its probability with respect to the class $y_i$:

$$p(y_i|\boldsymbol{t}) = \frac{\exp(\text{sim}(\phi^T(\boldsymbol{t}), \phi^T(\{\boldsymbol{V}, \boldsymbol{c}_i\}))/\tau)}{\sum_{j=1}^{K} \exp(\text{sim}(\phi^T(\boldsymbol{t}), \phi^T(\{\boldsymbol{V}, \boldsymbol{c}_j\}))/\tau)}. \tag{5}$$

$\phi^T_{\boldsymbol{V}}(\boldsymbol{t}) = [p(y_1|\boldsymbol{t}), p(y_2|\boldsymbol{t}), ..., p(y_K|\boldsymbol{t})]^\top \in \mathbb{R}^K$ is the prediction probability distribution of the textual description. The goal of correctly classifying textual descriptions can be formulated as:

$$\mathcal{L}_t = \sum_{(\boldsymbol{t}, y) \in \mathcal{D}_{txt}} \ell(\phi^T_{\boldsymbol{V}}(\boldsymbol{t}), y), \tag{6}$$

where $\mathcal{D}_{txt}$ is a set of pairs of textual descriptions and their corresponding classes, which can be conveniently created using some existing textual templates without customized design or collection.

The effectiveness of textual supervision in the context of test-time prompt learning stems from the following two aspects. Firstly, since image and text features generated by CLIP are aligned, correctly classifying class-related textual descriptions helps improve the visual discriminative capacity of the learned prompt. Moreover, unlike the pseudo-labels of test images, the labels of textual descriptions are accurately assigned according to the class names and thus provide noise-free supervision information for prompt learning.

**Diversity regularization.** Although we discard samples with small confidence scores, some of the selected samples are still attached with false pseudo-labels. We find that blindly trusting the pseudo-labels may harm the prompt learning, *e.g.*, resulting in a trivial solution, where the model classifies almost all samples into the same class (as demonstrated in Section 4.5). To solve this problem, we use a regularization term to diversify the model predictions:

$$\mathcal{L}_r = -\frac{|\mathcal{B}_{test}|}{Q} \mathbf{H}(\bar{\bar{\Phi}}_{\boldsymbol{V}}(\mathcal{B}_{test})), \tag{7}$$

where $\bar{\bar{\Phi}}_{\boldsymbol{V}}(\mathcal{B}_{test}) = \frac{1}{2Q} \sum_{\boldsymbol{x} \in \mathcal{B}_{test}} \mathbb{I}_{[\max(\Phi(\boldsymbol{x})) > \eta]}(\Phi_{\boldsymbol{V}}(\boldsymbol{x}) + \Phi_{\boldsymbol{V}}(\mathcal{A}(\boldsymbol{x})))$ is the averaged prediction probability distribution of the selected samples without and with strong data augmentations. The coefficient $\frac{|\mathcal{B}_{\text{test}}|}{Q}$ adaptively adjusts the magnitude of the regularization term. The value increases when fewer samples are selected; otherwise, decreases. The rationality is that fewer confident samples

imply a larger gap between the pre-training and the downstream task and thus our method relies more on the regularization term to alleviate the effect of false pseudo-labels, and vice versa.

With two balancing hyper-parameters $\lambda_t$ and $\lambda_r$, the task-oriented prompt $\boldsymbol{V}^{Task}$ can be obtained by minimizing the overall objective, defined as:

$$\boldsymbol{V}^{Task} = \arg \min_{\boldsymbol{V}} \mathcal{L}_p + \lambda_t \mathcal{L}_t + \lambda_r \mathcal{L}_r. \tag{8}$$

### 3.3 INSTANCE-ORIENTED PROMPT REFINEMENT

In the second stage of our TIPPLE, we learn the instance-oriented prompt. Different from TPT (Shu et al., 2022) which trains the prompt from the hand-craft template initialization, we propose to refine the learned task-oriented prompt for each test sample with one-step optimization. Based on the entropy minimization loss defined in Eq. (3) and the task-oriented prompt $\boldsymbol{V}^{Task}$, TIPPLE yields the prompt $\boldsymbol{V}^*$ for a given test sample $\boldsymbol{x}_{test}$:

$$\boldsymbol{V}^* = \boldsymbol{V}^{Task} + \boldsymbol{V}^{Inst}, \text{ where } \boldsymbol{V}^{Inst} = -\alpha \frac{\partial \mathbf{H}(\tilde{\Phi}_{\boldsymbol{V}}(\boldsymbol{x}_{test}))}{\partial \boldsymbol{V}} , \tag{9}$$

where $\alpha$ is the learning rate and $\boldsymbol{V}^{Inst}$ is called instance-oriented prompt residual. Utilizing the prompt $\boldsymbol{V}^*$, one can obtain the predication probability of the test sample $\boldsymbol{x}_{test}$.

### 3.4 EXTENSIONS

In the previous sections, we tackle a setting where the prompt is trained and evaluated on an unlabeled test dataset, which is entirely provided in advance. Benefiting from separately handling each batch of test samples (as illustrated in Eq. (4) and Eq. (7)) and learning the task-oriented prompt, our TIPPLE can be naturally applied to two additional scenarios as follows.

**Prompt learning on the online streaming data.** In practice, when adapting a pre-trained model to a specific downstream task, the unlabeled test data may arrive in an online stream and each batch can only be observed once (Wang et al., 2021a; Iwasawa & Matsuo, 2021; Liang et al., 2023). Besides, the model should immediately make predictions on the streaming data after the online optimization. We extend our method to this setting by optimizing the prompt on the given batch of test samples with the objective function defined in Eq. (8). After only a one-step back-propagation, we predict their labels using the learned prompt. Note that we continuously train the prompt based on that learned from previous test batches. Unlike TPT, which necessitates a one-step back-propagation for each test sample, our approach processes a batch of test samples simultaneously, leading to higher efficiency.

**Prompt learning on the unlabeled training data.** We also consider another scenario where a set of unlabeled samples (referred to as the unlabeled training data later) is available prior to testing. We expect that the prompt trained on the unlabeled training data can directly enhance the performance of CLIP on unseen test samples, even without test-time learning. This can not be achieved by TPT since it learns prompts on the fly with a single unlabeled test sample. Fortunately, the first stage of our TIPPLE enables prompt learning on the unlabeled training data, resulting in the task-oriented prompt, which shows strong generalization to unseen test samples as demonstrated in our experiments.

## 4 EXPERIMENTS

### 4.1 EVALUATION ON THE DATASETS FROM VARIOUS DOMAINS

**Benchmark datasets.** In this section, we conduct experiments on the 10 datasets from various domains. These datasets cover diverse recognition tasks including classification on generic objects (Caltech101 (Fei-Fei et al., 2004)), fine-grained classification (Flowers102 (Nilsback & Zisserman, 2008), OxfordPets (Parkhi et al., 2012), StanfordCars (Krause et al., 2013), Food101 (Bossard et al., 2014), FGVCAircraft (Maji et al., 2013)), texture classification (DTD (Cimpoi et al., 2014)), action recognition (UCF101 (Soomro et al., 2012)), scene classification (SUN397 (Xiao et al., 2010)), and satellite imagery recognition (EuroSAT (Helber et al., 2019)).

**Setup.** We compare our method with the zero-shot CLIP (Radford et al., 2021), few-shot prompt learning methods (CoOp (Zhou et al., 2022b) and CoCoOp (Zhou et al., 2022a)), and test-time prompt learning methods (TPT (Shu et al., 2022) and DiffTPT (Feng et al., 2023)). We also follow (Shu et al.,

Table 1: **Comparison on the datasets from various domains.** CLIP uses the default prompt and the ensemble of hand-crafted prompts in the zero-shot setting. CoOp and CoCoOp are trained on ImageNet with 16 labeled training samples per class. $CoOp_{PL}$ denotes that CoOp learns the prompt on the test set using our proposed pseudo-label generation method. We report the top-1 accuracy.

| Method | Flower102 | DTD | Pets | Cars | UCF101 | Caltech101 | Food101 | SUN397 | Aircraft | EuroSAT | Average |
|---|---|---|---|---|---|---|---|---|---|---|---|
| CLIP-RN50 (Radford et al., 2021) | 61.75 | 40.37 | 83.57 | 55.70 | 58.84 | 85.88 | 73.97 | 58.80 | 15.66 | 23.69 | 55.82 |
| Ensemble (Radford et al., 2021) | 62.77 | 40.37 | 82.97 | 55.89 | 59.48 | 87.26 | 74.82 | 60.85 | 16.11 | 25.79 | 56.63 |
| CoOp (Zhou et al., 2022b) | 61.55 | 37.29 | 87.00 | 55.32 | 59.05 | 86.53 | 75.59 | 58.15 | 15.12 | 26.20 | 56.18 |
| CoCoOp (Zhou et al., 2022a) | 65.57 | 38.53 | 88.39 | 56.22 | 57.10 | 87.38 | 76.20 | 59.61 | 14.61 | 28.73 | 57.23 |
| TPT (Shu et al., 2022) | 62.69 | 40.84 | 84.49 | 58.46 | 60.82 | 87.02 | 74.88 | 61.46 | 17.58 | 28.33 | 57.66 |
| $CoOp_{PL}$+TPT (Shu et al., 2022) | 65.25 | 37.65 | 86.78 | 30.23 | 59.87 | 88.52 | **77.54** | 57.09 | 16.08 | 9.33 | 52.83 |
| TIPPLE (Ours) | **65.61** | **44.25** | **89.87** | **58.89** | **63.82** | **89.02** | 77.50 | **63.13** | **18.33** | **35.68** | **60.61** |
| CLIP-ViT-B/16 (Radford et al., 2021) | 67.44 | 44.27 | 88.25 | 65.48 | 65.13 | 93.35 | 83.65 | 62.59 | 23.67 | 42.01 | 63.58 |
| Ensemble (Radford et al., 2021) | 66.99 | 45.04 | 86.92 | 66.11 | 65.16 | 93.55 | 82.86 | 65.63 | 23.22 | 50.42 | 64.59 |
| CoOp (Zhou et al., 2022b) | 68.71 | 41.92 | 89.14 | 64.51 | 66.55 | 93.70 | 85.30 | 64.15 | 18.47 | 46.39 | 63.88 |
| CoCoOp (Zhou et al., 2022a) | 70.85 | 45.45 | **90.46** | 64.90 | 68.44 | 93.79 | 83.97 | 66.89 | 22.29 | 39.23 | 64.63 |
| TPT (Shu et al., 2022) | 68.98 | 47.75 | 87.79 | 66.87 | 68.04 | 94.16 | 84.67 | 65.50 | 24.78 | 42.44 | 65.10 |
| $CoOp_{PL}$+TPT (Shu et al., 2022) | 70.81 | 47.05 | 89.37 | 56.45 | 66.46 | **94.69** | 85.77 | 67.21 | 23.04 | 28.33 | 62.92 |
| TIPPLE (Ours) | **71.30** | **49.17** | 90.15 | **67.80** | **71.25** | 93.94 | **86.01** | **68.13** | **25.36** | **51.77** | **67.49** |

Table 2: **Comparison on ImageNet and the OOD Datasets.** CLIP uses the default prompt and the ensemble of hand-crafted prompts in the zero-shot setting. CoOp and CoCoOp are trained on ImageNet with 16 labeled training samples per class. TPT and our TIPPLE are the test-time prompt learning methods. We report the top-1 accuracy.

| Method | ImageNet | ImageNet-A | ImageNet-V2 | ImageNet-R | ImageNet-Sketch | Average | OOD Average |
|---|---|---|---|---|---|---|---|
| CLIP-RN50 (Radford et al., 2021) | 58.16 | 21.83 | 51.41 | 56.15 | 33.37 | 44.18 | 40.69 |
| Ensemble (Radford et al., 2021) | 59.81 | 23.24 | 52.91 | 60.72 | 35.48 | 46.43 | 43.09 |
| CoOp (Zhou et al., 2022b) | **63.33** | 23.06 | 55.40 | 56.60 | 34.67 | 46.61 | 42.43 |
| CoCoOp (Zhou et al., 2022a) | 62.81 | 23.32 | **55.72** | 57.74 | 34.48 | 46.81 | 42.82 |
| TPT (Shu et al., 2022) | 60.74 | 26.67 | 54.70 | 59.11 | 35.09 | 47.26 | 43.89 |
| TIPPLE (Ours) | 62.73 | **29.13** | 55.49 | **64.17** | **38.49** | **50.00** | **46.82** |
| CLIP-ViT-B/16 (Radford et al., 2021) | 66.73 | 47.87 | 60.86 | 73.98 | 46.09 | 59.11 | 57.20 |
| Ensemble (Radford et al., 2021) | 68.34 | 49.89 | 61.88 | 77.65 | 48.24 | 61.20 | 59.42 |
| CoOp (Zhou et al., 2022b) | **71.51** | 49.71 | 64.20 | 75.21 | 47.99 | 61.72 | 59.28 |
| CoCoOp (Zhou et al., 2022a) | 71.02 | 50.63 | 64.07 | 76.18 | 48.75 | 62.13 | 59.91 |
| TPT (Shu et al., 2022) | 68.98 | 54.77 | 63.45 | 77.06 | 47.94 | 62.44 | 60.81 |
| TIPPLE (Ours) | 71.03 | **57.56** | **64.39** | **80.37** | **50.10** | **64.69** | **63.11** |

2022) to report the results of $CoOp_{PL}$+TPT for a comprehensive comparison, where CoOp is trained on the test set using our proposed pseudo-label generation method. All methods are evaluated with two visual backbones of CLIP, *i.e.*, ResNet-50 (RN50) (He et al., 2016) and ViT-B/16 (Dosovitskiy et al., 2020). More implementation details of baselines and our method can be found in Appendix B.

**Results.** Table 1 shows the comparison of different methods on 10 datasets. It shows that on some datasets, even though CoOp with pseudo-labels can improve the performance of TPT, it is inferior to our TIPPLE. On other datasets, we can observe that CoOp fails to advance TPT, indicating that it is infeasible to capture task-level knowledge using CoOp at test time. TIPPLE achieves higher accuracy than TPT on most of the datasets. The averaged performance gains are 2.95% and 2.39% on ResNet-50 and ViT-B/16 visual backbones, respectively. As shown in Appendix D, our method outperforms the state-of-the-art method, DiffTPT. For example, compared to DiffTPT, the averaged improvement on 7 datasets of TIPLLE is 1.87% on ViT-B/16 visual backbone. Besides, DiffTPT is more time-consuming and memory-heavy than ours due to its adopted diffusion models. As reported in Appendix G, DiffTPT consumes 441.1x time and 3.2x memory than our TIPPLE on average. These results verify the effectiveness of our TIPLLE on diverse downstream recognition tasks.

## 4.2 EVALUATION ON IMAGENET AND ITS OOD VARIANTS

**Benchmark datasets.** We conduct experiments on ImageNet (Deng et al., 2009) and evaluate the model robustness against natural distribution shifts on its out-of-distribution (OOD) variants containing different types of domain-shifted data. Sepcifically, we use four datasets: ImageNet-A (Hendrycks et al., 2021b), ImageNet-V2 (Recht et al., 2019), ImageNet-R (Hendrycks et al., 2021a), and ImageNet-Sketch(Wang et al., 2019). A brief overview of these datasets and the mplementation details can be found in Appendix C and Appendix B, respectively.

Table 3: **Cross-task evaluation of the proposed TIPPLE.** We learn the task-oriented prompt on ImageNet and apply it to 10 target datasets from various domains. The marker "◊" denotes testing without the instance-oriented prompt refinement. We report the top-1 accuracy.

| Method | Flower102 | DTD | Pets | Cars | UCF101 | Caltech101 | Food101 | SUN397 | Aircraft | EuroSAT | Average |
|---|---|---|---|---|---|---|---|---|---|---|---|
| CLIP-RN50 (Radford et al., 2021) | 61.75 | 40.37 | 83.57 | **55.70** | 58.84 | 85.88 | 73.97 | 58.80 | 15.66 | 23.69 | 55.82 |
| TIPPLE◊ (Ours) | **62.44** | **40.54** | **85.66** | 53.56 | **59.85** | **89.01** | **74.46** | **61.00** | **16.14** | **28.80** | **57.14** |
| TPT (Shu et al., 2022) | 62.69 | 40.84 | 84.49 | **58.46** | 60.82 | 87.02 | 74.88 | 61.46 | **17.58** | 28.33 | 57.66 |
| TIPPLE (Ours) | **62.81** | **41.90** | **86.89** | 56.47 | **60.93** | **89.49** | **75.59** | **62.63** | 16.47 | **30.38** | **58.36** |

Table 4: **Comparison in the scenario of the online streaming data.** CLIP uses the default prompt in the zero-shot setting. TPT learns a prompt for each test sample. Our TIPPLE-S processes a batch of test samples simultaneously, with batch sizes of 64, 128, and 256. We report the top-1 accuracy.

| Method | Flower102 | DTD | Pets | Cars | UCF101 | Caltech101 | Food101 | SUN397 | Aircraft | EuroSAT | Average |
|---|---|---|---|---|---|---|---|---|---|---|---|
| CLIP-RN50 (Radford et al., 2021) | 61.75 | 40.37 | 83.57 | 55.7 | 58.84 | 85.88 | 73.97 | 58.80 | 15.66 | 23.69 | 55.82 |
| TPT (Shu et al., 2022) | 62.69 | 40.84 | 84.49 | **58.46** | 60.82 | **87.02** | 74.88 | **61.46** | 17.58 | 28.33 | 57.66 |
| TIPPLE-S-64 (Ours) | **64.50** | **42.75** | 87.05 | 54.96 | **62.72** | 86.38 | 76.66 | 60.69 | 17.27 | 33.95 | 58.69 |
| TIPPLE-S-128 (Ours) | 64.35 | 42.59 | **88.03** | 56.06 | 62.41 | 86.96 | 76.78 | 60.94 | **17.93** | **34.77** | **59.08** |
| TIPPLE-S-256 (Ours) | 63.78 | 42.02 | 87.74 | 55.96 | 61.79 | 86.77 | **76.91** | 61.36 | 17.74 | 32.81 | 58.69 |

Table 5: **Comparison in the scenario where the unlabeled training data is available.** CLIP uses the default prompt in the zero-shot setting. TPT learns a prompt for each test sample. Our TIPPLE-T learns a task-oriented prompt on the unlabeled training data and uses it on test samples. The marker "◊" denotes testing without the instance-oriented prompt refinement. We report the top-1 accuracy.

| Method | Flower102 | DTD | Pets | Cars | UCF101 | Caltech101 | Food101 | SUN397 | Aircraft | EuroSAT | Average |
|---|---|---|---|---|---|---|---|---|---|---|---|
| CLIP-RN50 (Radford et al., 2021) | 60.58 | 39.09 | 83.92 | 55.55 | 58.49 | 85.59 | 73.47 | 58.51 | 15.00 | 24.24 | 55.44 |
| TPT (Shu et al., 2022) | 62.45 | 41.25 | 84.83 | **58.60** | 59.57 | **86.13** | 74.70 | 61.35 | 17.38 | 28.48 | 57.47 |
| TIPPLE-T◊ (Ours) | 63.63 | 41.96 | 85.26 | 55.87 | 61.01 | 86.04 | 76.89 | 61.25 | 17.71 | 32.12 | 58.17 |
| TIPPLE-T (Ours) | **64.53** | **42.63** | **87.97** | 56.13 | **62.65** | 85.90 | **77.25** | **61.43** | **18.63** | **33.87** | **59.10** |

**Results.** Table 2 shows the results of different methods on ImageNet and its OOD variants. It is surprising that TIPPLE achieves comparable performance with CoCoOp on ImageNet, which is trained with labeled training samples from ImageNet. On the OOD datasets, TIPPLE outperforms compared methods by a large margin, showing strong robustness against natural distribution shifts. The improvements of TIPPLE over TPT are 2.74% and 2.25% on average on ResNet-50 and ViT-B/16 visual backbones, respectively. Our results suggest that the task-level knowledge is helpful to the test-time prompt learning, especially for enhancing the model robustness to the OOD data.

## 4.3 CROSS-TASK EVALUATION

We evaluate the proposed TIPPLE in the cross-task setting. Specifically, following the cross-task setting in previous works (Zhou et al., 2022a; Zhu et al., 2022), we learn the task-oriented prompt on ImageNet and apply it to 10 target datasets from various domains. Compared to the original CLIP, only using the task-oriented prompt learned on ImageNet can significantly improve the performance on target datasets, indicating that TIPPLE helps the generalizability of the prompt and does not overfit the specific dataset. Furthermore, in this setting, our TIPPLE outperforms TPT on most datasets, where the gains are greater than 1% on 7 out of 10 datasets. These results demonstrate that the task-level prior knowledge utilized by our TIPPLE is transferable across different datasets.

## 4.4 RESULTS OF TWO EXTENSIONS

**Prompt learning on the online streaming data.** As illustrated in Section 3.4, we extend our TIPPLE to the scenario of the online streaming data, denoted as TIPPLE-S. Table 4 shows the results of zero-shot CLIP, TPT which processes test samples one by one, and our method with varying batch sizes. Our method with different batch sizes can surpass TPT in terms of average accuracy. In this setting, due to the fixed number of test samples, a larger batch size corresponds to less number of updates for our method. When the batch size is set to 128, TIPPLE-S achieves the best performance, since there is a nice trade-off between the number of updates and the stability of gradients.

**Prompt learning on the unlabeled training data.** For evaluation in this setting, we equally split each original test dataset into two groups, one as the unlabeled training dataset and the other as the test dataset. Our method (TIPPLE-T) learns the task-oriented prompt on the unlabeled training dataset and applies it to the test samples. The results are shown in Table 5. We can see that compared to

Table 6: **Effect of the diversity regularization loss $\mathcal{L}_r$.** The marker "$\diamond$" denotes testing without the instance-oriented prompt refinement. We report the top-1 accuracy.

| Method | ImageNet | ImageNet-A | ImageNet-V2 | ImageNet-R | ImageNet-Sketch | Average | OOD Average |
|---|---|---|---|---|---|---|---|
| CLIP-RN50 (Radford et al., 2021) | 58.16 | 21.83 | 51.41 | 56.15 | 33.37 | 44.18 | 40.69 |
| TIPPLE$^\diamond$ w/o $\mathcal{L}_r$ | 60.27 | 24.67 | 52.06 | 61.11 | 0.25 | 39.67 | 34.52 |
| TIPPLE$^\diamond$ w/ $\mathcal{L}_r$ | 61.06 | 25.32 | 53.11 | 62.37 | 36.77 | 47.73 | 44.39 |
| TIPPLE w/o $\mathcal{L}_r$ | 61.77 | 27.61 | 54.03 | 63.20 | 21.70 | 45.66 | 41.64 |
| TIPPLE w/ $\mathcal{L}_r$ | **62.73** | **29.13** | **55.49** | **64.17** | **38.49** | **50.00** | **46.82** |

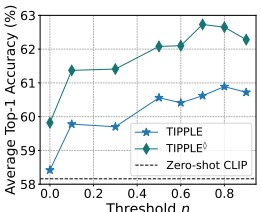 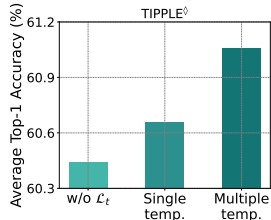 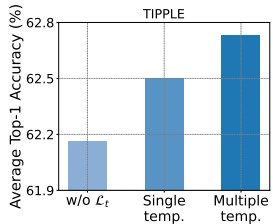

(a) Effect of the confidence threshold $\eta$      (b) Effect of the textual supervision

Figure 3: **Ablation on the confidence threshold $\eta$ (a) and the textual supervision (b).** In figure (b), w/o $\mathcal{L}_t$, using a single template, and using multiple templates are compared. The marker "$\diamond$" denotes testing without the instance-oriented prompt refinement. We evaluate with CLIP-RN50 on ImageNet.

zero-shot CLIP and TPT, our method achieve better performance even without the instance-oriented prompt refinement on the test samples, demonstrating the strong generalization of our task-oriented prompt. The two-stage TIPPLE-T shows a greater gain over TPT, *i.e.*, 1.63% on average.

### 4.5 ABLATION STUDIES

We investigate the effects of the confidence threshold $\eta$, the textual supervision, and the diversity regularization in this section. We provide more results and details in Appendix H.

**Effect of the confidence threshold $\eta$.** We filter the potential noisy pseudo-labels using the confidence threshold $\eta$. We investigate the effect of $\eta$ in Figure 3 (a). When $\eta$ is set as 0, *i.e.*, using all pseudo-labels, the performance is very poor, indicating the importance of selecting confident samples. We can see that TIPPLE can achieve large improvements than zero-shot CLIP when $\eta$ is greater than 0.5.

**Effect of the textual supervision.** We study the effect of the textual supervision by comparing the method without $\mathcal{L}_t$, creating textual descriptions with a single template "a photo of {class}", and creating textual descriptions with multiple templates as default. The results in Figure 3 (b) show that using the textual supervision can improve the performance of our method. Also, multiple templates provide more useful supervision information than a single one.

**Effect of the diversity regularization.** The results of our method without and with the diversity regularization loss are presented in Table 6. It shows that the diversity regularization can consistently improve the performance in all cases. Especially on ImageNet-Sketch, the method without the diversity regularization loss collapses, resulting in a trivial solution, where about 98% samples are classified into the same class. Therefore, the diversity regularization is an essential component in our method to bring performance improvements and avoid trivial solutions.

### 5 CONCLUSION

In this paper, we study the test-time prompt learning on VLMs. Our method TIPPLE improves existing methods which separately learn a prompt for each test sample. A two-stage training scheme is proposed to leverage both task- and instance-level knowledge. We learn the task-oriented prompt in the first stage and perform instance-oriented prompt refinement in the second stage. Extensive experiments verify the effectiveness of TIPPLE on various datasets.

**Limitations.** While TIPPLE does not require the labeled training data, due to the test-time learning, our method has an increased running time compared to zero-shot CLIP during testing. Besides, we adopt batch-wise training in the first stage and optimize with multiple augmented views of each test sample in the second stage, thus resulting in increased memory cost.

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

## A  BROADER IMPACT

The presented research should be categorized as research in the field of the adaptation of vision-language foundation models. Since our method does not require the labeled training data, it can be applied to scenarios where previous methods may be not feasible. We also believe that our work may inspire future studies to develop test-time prompt learning methods for large-scale foundation models in diverse downstream tasks, which reduce the cost of data annotations. Our method is built upon the pre-trained vision-language model, CLIP. However, as CLIP exhibits some unwanted biases as suggested in (Agarwal et al., 2021), our model may inherit these biases.

## B  IMPLEMENTATION DETAILS

In the zero-shot setting, we adopt the default prompt "a photo of a {class}" and the ensemble of 80 hand-crafted prompts as suggested in (Radford et al., 2021). Using a labeled training dataset, CoOp tunes a fixed prompt while CoCoOp trains a prompt generator conditioned on the image features. Following their original papers, we train both methods on ImageNet using 16 labeled training samples per class with 4 learnable context tokens and evaluate the learned prompt on all datasets. TPT learns a prompt for each test sample using its multiple augmented views. DiffTPT improves TPT by leveraging diffusion models to generate diverse augmented data. For TPT and DiffTPT, we adopt the settings from their original papers.

For our TIPPLE, we set the number of context tokens $M$ as 4. To learn the task-oriented prompt, we use the hand-crafted template "a photo of a {class}" as the initialization and train for 3 epochs with a batch size of 256, the AdamW optimizer, and an initial learning rate of 0.001, which is decayed with the factor 0.1 at every epoch. AugMix (Hendrycks et al., 2020) and Cutout (DeVries & Taylor, 2017) are included as strong data augmentations for optimizing the prompt. We create the set of textual descriptions using the default template "a photo of a {class}" and 7 extra templates. The 7 extra templates are selected for ImageNet series datasets in (Radford et al., 2021): "itap of a {class}.", "a bad photo of the {class}.", "a origami {class}.", "a photo of the large {class}.", "a {class} in a video game.", "art of the {class}.", and "a photo of the small {class}.". The default value of the confidence threshold $\eta$ is 0.7. The balancing parameter $\lambda_t$ is set as 0.1. $\lambda_r$ is set as 0.1 for the 4 OOD datasets and 0.2 for other 11 datasets. In the stage of the instance-oriented prompt refinement, as suggested in TPT, we use 64 augmented views and select the top 10% ($\rho$=0.1) confident samples. The AdamW optimizer is also employed with a learning rate of 0.001. All experimental results are averaged over three random seeds.

Table 7: **Comparison of zero-shot CLIP, DiffTPT, and our TIPPLE on the datasets from various domains.** We report the top-1 accuracy.

| Method | Flower102 | DTD | Pets | Cars | UCF101 | Caltech101 | Aircraft | EuroSAT | Average |
|---|---|---|---|---|---|---|---|---|---|
| CLIP-RN50 (Radford et al., 2021) | 61.75 | 40.37 | 83.57 | 55.70 | 58.84 | 85.88 | 15.66 | 23.69 | 53.18 |
| DiffTPT (Feng et al., 2023) | 63.22 | 41.31 | 85.12 | **59.33** | 63.20 | **89.70** | 18.25 | **41.70** | 57.73 |
| TIPPLE (Ours) | **65.61** | **44.25** | **89.87** | 58.89 | **63.82** | 89.02 | **18.33** | 35.68 | **58.18** |
| CLIP-ViT-B/16 (Radford et al., 2021) | 67.44 | 44.27 | 88.25 | 65.48 | 65.13 | 93.35 | 23.67 | 42.01 | 61.2 |
| DiffTPT (Feng et al., 2023) | 69.47 | 47.34 | 87.95 | 67.45 | 68.68 | **94.69** | 24.96 | 45.20 | 63.22 |
| TIPPLE (Ours) | **71.30** | **49.17** | **90.15** | **67.80** | **71.25** | 93.94 | **25.36** | **51.77** | **65.09** |

Table 8: **Comparison on the datasets from various domains.** CLIP uses the default prompt in the zero-shot setting. CoOp and CoCoOp are trained on ImageNet with 16 labeled training samples per class. TPT and our TIPPLE are the test-time prompt learning methods. We report the top-1 accuracy with the standard deviation. Note that the results of the baseline methods (CLIP-RN50, CoOp, CoCoOp, and TPT) are drawn from Section A.2 in (Shu et al., 2022).

| Method | Flower102 | DTD | Pets | Cars | UCF101 | Caltech101 | Food101 | SUN397 | Aircraft | EuroSAT | Average |
|---|---|---|---|---|---|---|---|---|---|---|---|
| CLIP-RN50 (Radford et al., 2021) | 61.75 | 40.37 | 83.57 | 55.70 | 58.84 | 85.88 | 73.97 | 58.80 | 15.66 | 23.69 | 55.82 |
| CoOp (Zhou et al., 2022b) | 61.62 ($\pm$ 0.2) | 37.77 ($\pm$ 0.9) | 87.24 ($\pm$ 0.2) | 55.72 ($\pm$ 0.8) | 59.89 ($\pm$ 0.8) | 87.23 ($\pm$ 0.6) | 75.86 ($\pm$ 0.2) | 59.28 ($\pm$ 0.9) | 15.20 ($\pm$ 0.4) | 25.43 ($\pm$ 4.0) | 56.52 ($\pm$ 0.7) |
| CoCoOp (Zhou et al., 2022a) | 65.11 ($\pm$ 1.0) | 39.14 ($\pm$ 0.7) | 87.83 ($\pm$ 0.6) | 56.40 ($\pm$ 0.3) | 58.57 ($\pm$ 1.0) | 86.95 ($\pm$ 0.5) | 76.18 ($\pm$ 0.5) | 60.62 ($\pm$ 0.9) | 15.13 ($\pm$ 0.5) | 28.79 ($\pm$ 0.9) | 57.47 ($\pm$ 0.2) |
| TPT (Shu et al., 2022) | 62.80 ($\pm$ 0.3) | 41.43 ($\pm$ 0.5) | 84.42 ($\pm$ 0.1) | 58.53 ($\pm$ 0.1) | 60.64 ($\pm$ 0.3) | 87.23 ($\pm$ 0.2) | 75.02 ($\pm$ 0.1) | 61.46 ($\pm$ 0.0) | 17.60 ($\pm$ 0.4) | 28.46 ($\pm$ 0.1) | 57.76 ($\pm$ 0.1) |
| TIPPLE (Ours) | **65.61** ($\pm$ 0.7) | **44.25** ($\pm$ 0.7) | **89.87** ($\pm$ 0.1) | **58.89** ($\pm$ 0.1) | **63.82** ($\pm$ 0.8) | **89.02** ($\pm$ 0.2) | **77.50** ($\pm$ 0.2) | **63.13** ($\pm$ 0.1) | **18.33** ($\pm$ 0.0) | **35.68** ($\pm$ 5.0) | **60.61** ($\pm$ 0.5) |
| CLIP-ViT-B/16 (Radford et al., 2021) | 67.44 | 44.27 | 88.25 | 65.48 | 65.13 | 93.35 | 83.65 | 62.59 | 23.67 | 42.01 | 63.58 |
| CoOp (Zhou et al., 2022b) | 68.25 ($\pm$ 0.5) | 42.34 ($\pm$ 2.0) | 89.38 ($\pm$ 0.2) | 63.35 ($\pm$ 1.0) | 67.17 ($\pm$ 1.0) | 92.82 ($\pm$ 0.5) | 83.74 ($\pm$ 0.4) | 64.51 ($\pm$ 0.6) | 19.99 ($\pm$ 2.0) | 40.22 ($\pm$ 4.0) | 63.18 ($\pm$ 0.7) |
| CoCoOp (Zhou et al., 2022a) | 71.59 ($\pm$ 0.6) | 45.48 ($\pm$ 0.2) | 90.20 ($\pm$ 0.2) | 65.17 ($\pm$ 0.2) | 68.77 ($\pm$ 0.8) | **94.15** ($\pm$ 0.3) | 84.83 ($\pm$ 1.0) | 67.07 ($\pm$ 0.3) | 22.95 ($\pm$ 0.7) | 42.13 ($\pm$ 3.0) | 65.23 ($\pm$ 0.6) |
| TPT (Shu et al., 2022) | 68.79 ($\pm$ 0.1) | 46.79 ($\pm$ 0.1) | 87.09 ($\pm$ 0.1) | 66.38 ($\pm$ 0.2) | 67.86 ($\pm$ 0.1) | 94.13 ($\pm$ 0.1) | 84.67 ($\pm$ 0.1) | 65.41 ($\pm$ 0.1) | 23.44 ($\pm$ 0.3) | 42.78 ($\pm$ 0.3) | 64.73 ($\pm$ 0.1) |
| TIPPLE (Ours) | **71.30** ($\pm$ 0.4) | **49.17** ($\pm$ 0.2) | **90.15** ($\pm$ 0.1) | **67.80** ($\pm$ 1.0) | **71.25** ($\pm$ 0.7) | 93.94 ($\pm$ 0.3) | **86.01** ($\pm$ 0.1) | **68.13** ($\pm$ 0.2) | **25.36** ($\pm$ 1.9) | **51.77** ($\pm$ 0.8) | **67.49** ($\pm$ 0.1) |

## C   A Brief Overview of the OOD Datasets

A brief overview of the OOD datasets is:

- *ImageNet-A* (Hendrycks et al., 2021b) contains 7,500 natural images of 200 ImageNet categories that are misclassified by a standard ResNet-50 (He et al., 2016);

- *ImageNet-V2* (Recht et al., 2019) is a newly collected version of the original ImageNet validation set. The dataset includes 10,000 natural images, covering 1,000 ImageNet categories;

- *ImageNet-R* (Hendrycks et al., 2021a) includes 30,000 images of 200 ImageNet categories with various artistic renditions, *e.g.*, paintings, embroidery;

- *ImageNet-Sketch* (Wang et al., 2019) contains sketch-like images and matches the ImageNet validation set in categories and scale.

## D   Comparison With the State-of-the-art: DiffTPT

Due to the huge inference cost of DiffTPT, the results on two large-scale test sets (Food101 and SUN397) are missing in its original paper. Hence, we compare our TIPPLE with the state-of-the-art method DiffTPT on the remaining 7 datasets in Table 7. We can see that our method outperforms DiffTPT in terms of top-1 accuracy. Specifically, compared to DiffTPT, the averaged improvements on 7 datasets of TIPLLE are 0.45% and 1.87% on ResNet-50 and ViT-B/16 visual backbones, respectively. Besides, DiffTPT is more time-consuming and memory-heavy than ours due to its adopted diffusion models. As reported in Appendix G, DiffTPT consumes 441.1x time and 3.2x memory than our TIPPLE on average, indicating it impractical in the real-world setting.

## E   Results with Error Bars

We report the top-1 accuracy with an error bar (standard deviation) in Tables 8 and 9. The average accuracy and standard deviation are calculated by running with three random seeds. Compared to TPT, TIPPLE achieves significant improvements in terms of the top-1 accuracy on most datasets. In addition, even though taking the error bars into consideration, our method can still outperform baseline methods by a large margin.

Table 9: **Comparison on ImageNet and the OOD Datasets.** CLIP uses the default prompt in the zero-shot setting. CoOp and CoCoOp are trained on ImageNet with 16 labeled training samples per class. TPT and our TIPPLE are the test-time prompt learning methods. We report the top-1 accuracy with the standard deviation. Note that the results of the baseline methods (CLIP-RN50, CoOp, CoCoOp, and TPT) are drawn from Section A.2 in (Shu et al., 2022).

| Method | ImageNet | ImageNet-A | ImageNet-V2 | ImageNet-R | ImageNet-Skech | Average | OOD Average |
|---|---|---|---|---|---|---|---|
| CLIP-RN50 (Radford et al., 2021) | 58.16 | 21.83 | 51.41 | 56.15 | 33.37 | 44.18 | 40.69 |
| CoOp (Zhou et al., 2022b) | **63.27** ($\pm$ 0.07) | 23.23 ($\pm$ 0.19) | 55.50 ($\pm$ 0.09) | 57.08 ($\pm$ 0.42) | 34.68 ($\pm$ 0.03) | 46.75 ($\pm$ 0.12) | 42.62 ($\pm$ 0.17) |
| CoCoOp (Zhou et al., 2022a) | 62.86 ($\pm$ 0.11) | 23.38 ($\pm$ 0.50) | **55.59** ($\pm$ 0.14) | 57.55 ($\pm$ 0.23) | 34.74 ($\pm$ 0.29) | 46.82 ($\pm$ 0.21) | 42.82 ($\pm$ 0.23) |
| TPT (Shu et al., 2022) | 60.77 ($\pm$ 0.03) | 26.60 ($\pm$ 0.13) | 54.70 ($\pm$ 0.11) | 59.08 ($\pm$ 0.03) | 35.17 ($\pm$ 0.08) | 47.27 ($\pm$ 0.00) | 43.89 ($\pm$ 0.00) |
| TIPPLE (Ours) | 62.73 ($\pm$ 0.08) | **29.13** ($\pm$ 0.19) | 55.49 ($\pm$ 0.19) | **64.17** ($\pm$ 0.18) | **38.49** ($\pm$ 0.28) | **50.00** ($\pm$ 0.11) | **46.82** ($\pm$ 0.16) |
| CLIP-ViT-B/16 (Radford et al., 2021) | 66.73 | 47.87 | 60.86 | 73.98 | 46.09 | 59.11 | 57.20 |
| CoOp (Zhou et al., 2022b) | **71.71** ($\pm$ 0.19) | 49.99 ($\pm$ 0.29) | **64.49** ($\pm$ 0.39) | 75.51 ($\pm$ 0.26) | 48.10 ($\pm$ 0.14) | 61.96 ($\pm$ 0.25) | 59.52 ($\pm$ 0.26) |
| CoCoOp (Zhou et al., 2022a) | 70.70 ($\pm$ 0.32) | 50.76 ($\pm$ 0.13) | 63.93 ($\pm$ 0.19) | 76.09 ($\pm$ 0.29) | 48.60 ($\pm$ 0.38) | 62.02 ($\pm$ 0.20) | 59.85 ($\pm$ 0.19) |
| TPT (Shu et al., 2022) | 68.96 ($\pm$ 0.03) | 54.47 ($\pm$ 0.26) | 63.46 ($\pm$ 0.07) | 77.10 ($\pm$ 0.04) | 47.93 ($\pm$ 0.03) | 62.38 ($\pm$ 0.05) | 60.74 ($\pm$ 0.06) |
| TIPPLE (Ours) | 71.03 ($\pm$ 0.38) | **57.56** ($\pm$ 0.28) | 64.39 ($\pm$ 0.22) | **80.37** ($\pm$ 0.23) | **50.10** ($\pm$ 0.26) | **64.69** ($\pm$ 0.07) | **63.11** ($\pm$ 0.06) |

Table 10: **Comparison in the scenario of the online streaming data on ImageNet and the OOD Datasets.** CLIP uses the default prompt in the zero-shot setting. TPT learns a prompt for each test sample. Our TIPPLE-S processes a batch of test samples simultaneously, with batch sizes of 64, 128, and 256. We report the top-1 accuracy.

| Method | ImageNet | ImageNet-A | ImageNet-V2 | ImageNet-R | ImageNet-Sketch | Average | OOD Average |
|---|---|---|---|---|---|---|---|
| CLIP-RN50 (Radford et al., 2021) | 58.16 | 21.83 | 51.41 | 56.15 | 33.37 | 44.18 | 40.69 |
| TPT (Shu et al., 2022) | 60.74 | **26.67** | **54.70** | 59.11 | 35.09 | **47.26** | **43.89** |
| TIPPLE-S-64 (Ours) | 60.06 | 22.53 | 52.78 | 61.19 | 34.85 | 46.28 | 42.84 |
| TIPPLE-S-128 (Ours) | 60.37 | 24.36 | 52.78 | 61.14 | 35.33 | 46.80 | 43.40 |
| TIPPLE-S-256 (Ours) | **60.86** | 24.22 | 52.97 | **61.31** | **36.49** | 47.17 | 43.75 |

## F  RESULTS OF TWO EXTENSIONS

**Implementation details.** We conduct experiments with the ResNet-50 visual backbone to evaluate two extensions of our proposed TIPPLE described in Section 3.4. In the scenario of the online streaming data, we decrease the balancing hyper-parameter $\lambda_r$ and keep other settings unchanged as those stated in Appendix B. It is intuitive that overly diversifying the model predictions may hurt the performance when the number of selected samples is small, especially for datasets with a large number of classes. Therefore, we decay the default $\lambda_r$ by the factor 0.5 on SUN397, ImageNet, and ImageNet-Sketch for small batch sizes (64 and 128). In the scenario where the unlabeled training data is available, we equally split each original test dataset into two groups, one as the unlabeled training dataset and the other as the test dataset. Our method learns the task-oriented prompt on the unlabeled training dataset and applies it to the test samples. In this scenario, we adopt all default settings illustrated in Appendix B.

**Results.** We report the results on 10 datasets from different domains in Tables 4 and 5. The results on ImageNet and the OOD datasets are reported here. Table 10 shows that our method surpasses TPT on 3 datasets among 5, indicating the superiority of our method in handling the streaming data. Besides, we would like to emphasize that our method has a higher efficiency than TPT, since TIPPLE processes a batch of test samples simultaneously while TPT requires a one-step back-propagation for each test sample. We can see from Table 11 that TIPPLE achieves better performance than baselines in all cases. These results further confirm the strong generalization of our task-oriented prompt.

## G  INFERENCE COST

We report the averaged memory cost and averaged inference time per sample of different methods in Table 12. We evaluate these method on a single GeForce RTX 3090 GPU. We can see that, compared to TPT, the increased memory cost of TIPPLE is marginal, since both adopt batch-wise training. Also, because TIPPLE learns the task-oriented prompt with a very small number of epochs (3 epochs in our experiments), the increased inference time of TIPPLE is acceptable. In short, our TIPPLE

Table 11: **Comparison in the scenario where the unlabeled training data is available on ImageNet and the OOD Datasetss.** CLIP uses the default prompt in the zero-shot setting. TPT learns a prompt for each test sample. Our TIPPLE-T learns a task-oriented prompt on the unlabeled training data and uses it on test samples. The marker "$\diamond$" denotes testing without the instance-oriented prompt refinement. We report the top-1 accuracy.

| Method | ImageNet | ImageNet-A | ImageNet-V2 | ImageNet-R | ImageNet-Sketch | Average | OOD Average |
|---|---|---|---|---|---|---|---|
| CLIP-RN50 (Radford et al., 2021) | 58.14 | 21.87 | 51.61 | 56.02 | 33.3 | 44.19 | 40.70 |
| TPT (Shu et al., 2022) | 60.69 | 26.52 | 54.51 | 58.94 | 35.18 | 47.17 | 43.79 |
| TIPPLE-T$^{\diamond}$ (Ours) | 61.11 | 24.76 | 53.31 | 61.68 | 36.89 | 47.55 | 44.16 |
| TIPPLE-T (Ours) | **62.82** | **27.19** | **55.65** | **63.76** | **38.72** | **49.63** | **46.33** |

Table 12: **Comparison of TPT, DiffTPT, our TIPPLE, and our TIPPLE-S in terms of the averaged memory cost and averaged inference time per sample.** Note that our TIPPLE-S processes a batch of test samples simultaneously, thus has the shortest inference time. We report the results averaged over 10 datasets from various domains.

| | TPT | DiffTPT | TIPPLE (Ours) | TIPPLE-S (Ours) |
|---|---|---|---|---|
| Averaged Memory Cost ($GB$) | 6.73 | 22.13 | 6.94 | 7.59 |
| Averaged Inference Time per Sample ($S$) | 1.08 | 555.78 | 1.26 | 0.05 |

brings 2.5% gains on average over TPT with a little extra cost. DiffTPT is more time-consuming and memory-heavy than other methods due to its adopted diffusion models. It consumes 441.1x time and 3.2x memory than our TIPPLE, which makes DiffTPT inapplicable in some practical scenarios.

We also evaluate one of our extensions, TIPPLE-S, which processes a batch of test samples simultaneously, leading to higher efficiency. Our results verify that compared to TPT, TIPPLE-S achieves about 25x inference speed on average. It is worth noting that in terms of top-1 accuracy, TIPPLE-S is better than TPT, as shown in Table 4.

## H  ABLATION STUDIES

**Implementation details.** To investigate the effect of the confidence threshold $\eta$ on our proposed method, we conduct experiments with $\eta \in \{0, 0.1, 0.3, 0.5, 0.6, 0.7, 0.8, 0.9\}$. Because a large $\eta$ corresponds to a small number of selected samples, we decrease the balancing hyper-parameter $\lambda_r$ to avoid overly diversifying the model predictions as illustrated in Section 3.4. Specifically, we decay the default $\eta$ by the factor 0.5 (for $\eta = 0.8$ on ImageNet and ImageNet-Sketch, $\eta = 0.9$ on ImageNet-A and Image-V2) and the factor 0.25 (for $\eta = 0.9$ on ImageNet and ImageNet-Sketch). We study the effect of the textual supervision by comparing TIPPLE without $\mathcal{L}_t$, using a single template, and using multiple templates. To ablate the contribution of the diversity regularization, we evaluate the baseline without $\mathcal{L}_r$ ($\lambda_r = 0$). We also study the effect of the value of the trade-off parameter $\lambda_r$ with $\lambda_r \in \{0, 0.01, 0.05, 0.1, 0.15, 0.2, 0.3\}$.

**Results.** Table 13 shows that when $\lambda_r \in [0.05, 0.15]$, the diversity regularization term has a significant positive effect on the final results. The rationale is that a smaller $\lambda_r$ cannot encourage the prediction diversity, while a larger value weakens learning from the pseudo-label and the text supervision. Because the $\lambda_r$ is effective in a large range and the performance is consistent across different datasets, it is easy to select the parameter $\lambda_r$ in practice.

## I  ANALYSE OF THE DIVERSITY REGULARIZATION

In this section, for a more comprehensive analyse, we study the diversity regularization loss on the imbalanced datasets.

**Original diversity regularization loss.** We first construct the imbalanced datasets. For each class $i$, we randomly select $|\mathcal{D}_{test}^i| = N \cdot i^{-\gamma} (\gamma > 0)$ samples from the original balanced dataset, where $N$ is the number of samples for each class of the original balanced dataset. The parameter $\gamma$ controls the imbalanced degree, where a larger value corresponds to a more imbalanced dataset, and vice versa. For instance, for a dataset containing 100 categories, when $\gamma$ is set as 1/2 and 1, $\max(|\mathcal{D}_{test}^i|)/\min(|\mathcal{D}_{test}^i|)$ is 10 and 100, respectively. We study the effect of the diversity

Table 13: **Ablation on the trade-off parameter of the diversity regularization** $\lambda_r$**.** We report the top-1 accuracy.

| $\lambda_r$ | ImageNet | ImageNet-A | ImageNet-V2 | ImageNet-R | ImageNet-Skecth | Average | Average OOD |
|---|---|---|---|---|---|---|---|
| 0 | 61.77 | 27.61 | 54.03 | 63.20 | 21.70 | 45.66 | 41.64 |
| 0.01 | 62.16 | 27.71 | 54.83 | 63.44 | 22.16 | 46.06 | 42.04 |
| 0.05 | 62.11 | 28.57 | 55.27 | 63.73 | 34.31 | 48.80 | 45.47 |
| 0.1 | 62.21 | 29.13 | 55.49 | 64.17 | 38.49 | 49.90 | 46.82 |
| 0.15 | 62.08 | **29.16** | **55.56** | 64.38 | **39.25** | **50.09** | **47.09** |
| 0.2 | **62.73** | 28.96 | 55.21 | 64.32 | 21.22 | 46.49 | 42.43 |
| 0.3 | 51.33 | 28.84 | 38.62 | **64.55** | 23.11 | 41.29 | 38.78 |

Table 14: **Effect of the diversity regularization on the imbalanced datasets.** The imbalanced dataset is sampled from its original version. The parameter $\gamma$ controls the imbalanced degree, where a larger value corresponds to a more imbalanced dataset, and vice versa. We report the top-1 accuracy.

| Method | $\gamma = \frac{1}{2}$ | | | | $\gamma = 1$ | | | |
|---|---|---|---|---|---|---|---|---|
| | Flowers102 | DTD | Pets | Average | Flowers102 | DTD | Pets | Average |
| CLIP-RN50 | 63.94 | 32.23 | 81.65 | 59.27 | 62.50 | 37.09 | 85.93 | 61.84 |
| w/o $\mathcal{L}_r$ | 64.82 | 36.73 | 82.61 | 61.39 | 62.50 | 39.07 | 87.41 | 62.99 |
| w/ $\mathcal{L}_r$ | 65.58 | 37.20 | 83.73 | 62.17 | 63.75 | 35.76 | 87.07 | 62.19 |
| w/ Re-weighted $\mathcal{L}_r$ | **66.83** | **37.92** | **85.46** | **63.40** | **64.38** | **41.06** | **88.56** | **64.67** |

regularization loss with $\gamma = 1/2$ and $\gamma = 1$, as shown in Table 14. The results show that when $\gamma = 1/2$, the diversity regularization loss shows a positive effect on different datasets. In severely imbalanced cases ($\gamma = 1$), it has a negative impact on prediction results on DTD and Pets. Our results illustrate the effectiveness of the diversity regularization loss on the mildly imbalanced datasets.

**Re-weighted diversity regularization loss.** When there is an assumption on the imbalanced label distribution, we design the re-weighted diversity regularization loss to further improve the performance on the imbalanced datasets. Specifically, let $p \in \mathbb{R}^K$ denote the averaged prediction probability distribution in Eq. 7, we re-weight the element $p_i$ with $p_i \leftarrow r^\omega \cdot p_i$, where $r$ denotes the index of $p_i$ in the ascendingly sorted probability vector $p$ and $\omega > 0$ is the hyper-parameter which is set as $\gamma$ for the sake of simplicity. As shown in Table 14, the re-weighted diversity regularization loss shows consistent improvements over the methods without the diversity regularization loss and with the original diversity regularization loss. Our results confirm that this strategy adaptively improves the probability values of minority classes, and thus helps apply the diversity regularization loss on the imbalanced datasets.

