# OpenReview forum: "Task-to-Instance Prompt Learning for Vision-Language Models at Test Time"
_ICLR.cc/2024/Conference — ICLR 2024 Conference Withdrawn Submission_

### Official Review · Reviewer_43Ye · 2023-10-30

**Soundness:** 3 good
**Presentation:** 3 good
**Contribution:** 3 good
**Rating:** 5
**Confidence:** 4

**Summary:**

Existing test-time prompt learning often separately learn a prompt for each test sample, neglecting task-level knowledge. This paper proposes to learn both task-level and instance-level knowledge with a two-stage training strategy. In the first stage, TIPPLE is trained on unlabeled test dataset to learn task-oriented prompt with visual/textual supervision and diversity regularization; in the second stage, a residual prompt is learned for each test sampling using entropy minimization. Experiments are conducted on diverse image classification tasks, demonstrating SOTA performances. Two additional scenarios, online streaming data and unlabeled training data, are studied.

**Strengths:**

+ The studied task of test-time prompt learning for vision-language models is important.
+ The proposed method learns both task-level and instance-level knowledge. The motivation is resonable. To effectively learn task-level knowledge, the authors proposed two components of textual supervision and diversity regularization.
+ Experiments on datasets from various domains and ImageNet show clear improvement in accuracies. TIPPLE also outperforms TPT on online streaming and unlabeled training data scenarios. Ablation studies are adequate.
+ Figure illustrations are presented in a nice way.

**Weaknesses:**

- My concern is on the setting of this paper and its comparison with previous works. The authors claim that existing test-time prompt learning methods neglect the task-level knowledge. However, the proposed method relies on the entire test samples, which may not be available in practical test-time adaptation task. In Sec. 3.4, they discuss the online streaming data task. A minimum batch size of 64 is used in the experiments. It is unclear if a much smaller minibatch would still suffice to learn task-level knowledge. In some sense, the comparison with TPT is unfair as the two papers are handling different settings.
- The task-oriented prompt learning stage combines a few techniques that may not be quite novel. Pseudo-labeling with strong augmentation is a common SSL technique. The loss term in Eq.(8) consists of three terms, and the two balancing hyper-parameters need to be adjusted per task. The decompostion of task-level and instance-level prompts is also straightforward.
- Mathematical notations (like $\Phi_V, \phi_V^T$) are a little confusing.

**Questions:**

- In 4.1, the authors mention 'CoOp is trained on the test set using .. pseudo-label generation method'. Is strong data augmentations also used in CoOp and other comparison methods?
- The diversity regularization seems to be critical to avoid degraded performances. From Table 6,13, an improper setting of $\lambda_r$ leads to poor accuracies on ImageNet-Sketch. And a small change (~0.05) in $\lambda_r$ results in very different results. Could the authors comment more on this, especially when the available test images are few, say in online streaming data scenario?

---

### Official Review · Reviewer_R3mK · 2023-10-30

**Soundness:** 3 good
**Presentation:** 3 good
**Contribution:** 3 good
**Rating:** 6
**Confidence:** 5

**Summary:**

The paper introduces a novel test-time prompt learning method for vision-language models called Task-to-Instance PromPt LEarning (TIPPLE). The method aims to learn prompts for downstream tasks using only unlabeled test data. TIPPLE leverages both task- and instance-level knowledge in a two-stage training strategy, achieving superior performance on various downstream datasets.

**Strengths:**

1) The paper addresses an important and practical research topic of test-time prompt learning for VLMs, and introduces a novel approach that leverages both task- and instance-level knowledge, which leads to improved performance compared to existing methods, like TPT.

2) The paper reformulates the effective online pseudo-labeling paradigm along with two tailored components and provides comprehensive experimental results demonstrating the effectiveness and superiority of various components.

Building on previous test-time prompt learning methods and extending them to task-level prompt learning, the authors propose three key components to achieve this goal. Based on weak novelty but extensive experimental results and well writing, I recommend weak accept for this paper.

**Weaknesses:**

This paper is written well and reproducible. Providing several components with limited technical contribution, the proposed method is with weak novelty.

1) The Textual supervision module seems similar to LASP[1], which both add a cross-entropy text-to-text loss that enforces the learned prompts to be close in embedding space for exploiting the intrinsic information captured by the text encoder. The authors need to clearly explain the differences between the two methods.

2) Instance-oriented prompt refinement is trivial since it mimics the way in TPT, just with different initialization parameters.

3) Compared to TPT, this method seems limited as it needs enough unlabeled test samples for task-oriented prompt learning. Even authors provide extensions for prompt learning on the online streaming data, the implementation of the scenario of the online streaming data is still with a big batch size of at least 64. Results with fewer test samples should be provided for the validation of the effectiveness of the proposed methods.

[1] LASP: Text-to-Text Optimization for Language-Aware Soft Prompting of Vision & Language Models

**Questions:**

see weakness

---

### Official Review · Reviewer_cDhi · 2023-10-30

**Soundness:** 3 good
**Presentation:** 3 good
**Contribution:** 3 good
**Rating:** 6
**Confidence:** 4

**Summary:**

This paper proposes a test-time prompt learning method, named Task-to-Instance PromPt LEarning (TIPPLE), using unlabeled test data for prompt learning. To address the problem of the neglected task-level knowledge, TIPPLE adopts a two-stage training strategy to leverage both task- and instance-level knowledge. During the task-level optimization, TIPPLE uses an auxiliary text classification task and a diversity regularization term for the training of unlabeled data. Experiments show superior performance on various downstream datasets.

**Strengths:**

This paper is dedicated to an interesting issue that each test sample may lack task-level knowledge. To address this issue, this paper proposes task-oriented prompt learning to well learn task-level knowledge using batch samples. This knowledge can be well transferred to the next instance-oriented prompt refinement stage. Experiments in Table 6 also shows the effectiveness of the task-oriented prompt learning.

**Weaknesses:**

1. Ablation on the textual supervision, i.e., Eq. (5), is missing. Table 6 shows the effectiveness of the the diversity regularization and the task-oriented prompt learning, however, it lacks the effect of the textual supervision. It would be better to ablate the textual supervision.
2. Is the task-oriented prompt learning sensitive to the selection of the batch-wise samples, i.e., the order of the batch samples?
3. How about learning the task-oriented prompt for longer? I noticed that the learning of the task-oriented prompt is 3 epoch by default. What will happen when learning longer or shorter? Will the performance drop?
4. Typos and more clarification of the notations. For example, i) what is $z$ in line 12 in page 4 (above Eq. (2))? Is it $x$ instead? ii) What is $L_p$, $L_t$ and $L_r$ in Figure 1 (c)? I understand after reading the Method Section, but it is confused when reading Introduction Section. iii) "datasetwith" should be "dataset with" in line 2 in page 2.

**Questions:**

Please refer to the Weaknesses Section.

---

### Official Review · Reviewer_GbQ6 · 2023-10-30

**Soundness:** 3 good
**Presentation:** 3 good
**Contribution:** 2 fair
**Rating:** 5
**Confidence:** 5

**Summary:**

The paper introduces a prompt-tuning method designed to adapt vision-language models, such as CLIP, to downstream image classification tasks without relying on annotated data. Rather than solely creating an instance-prompt for each test sample, the method innovatively incorporates an additional phase to learn a task-prompt using multiple unlabeled datasets. Three distinct self-supervised learning techniques are introduced: firstly, an online pseudo-labeling paradigm equipped with a manually-tuned threshold provides visual supervision. Secondly, an auxiliary text classification task offers textual supervision. Lastly, a specially crafted regularization term ensures prediction diversification. The method is further evaluated in two additional scenarios, including online streaming data and unlabeled training data. Its effectiveness is robustly confirmed across various benchmark datasets and through comprehensive ablation studies.

**Strengths:**

* The paper is well presented, facilitating an easy understanding of the research idea at both conceptual and technical levels.

* The author presents a thorough ablation study to underscore the effectiveness of each introduced component.

* Beyond conventional evaluation scenarios, the paper delves into two additional real-world settings: online streaming data and the use of unlabeled training data.

**Weaknesses:**

* The novelty of this work leans more towards engineering than fundamental research. The approach integrates an off-the-shelf stage for learning task prompts, in contrast to previous works that solely focus on learning instance prompts. Specifically, when drawing from unlabeled data, the author incorporates familiar techniques in self-supervised learning, such as pseudo-labeling and hand-crafted regularization. However, these techniques rely on hyperparameter tuning across various downstream datasets. In summary, while the evaluation underscores the method's efficacy, the proposed approach primarily combines existing techniques to address a well-established problem, offering limited novel insights. Therefore, I recommend that the author consider submitting the paper to more application-oriented venues, such as CVPR.

* Given that the key of the proposed approach revolves around generating high-quality pseudo-labels, the experimental section falls short in assessing self-supervision signals as the intermediate beyond mere final classification accuracy. For example, it would be valuable for the author to include both quantitative evaluations or qualitative visualizations to highlight how the proposed component minimizes the noise level and avoids falsely generated pseudo-labels. Additionally, it would be insightful to determine if visual and textual supervision would conflict and lead to poor embedding space.

**Questions:**

* I expect the author could provide more evidence to show the novelty beyond combining existing self-supervised techniques to make incremental improvements over the previous works.

* The author could provide more insights or experiments on how each proposed component improves the quality of pseudo-supervision signals. For example, how does poor visual supervision from a noisy pseudo-label affect the embedding space? How to determine whether the visual and textual supervision align well or contradict each other in embedding space? How does the regularity term improve the pseudo-labels? In sum, the improvement in the final classification accuracy can show the effectiveness of the proposed method but may hide some important insights.